# Identifying Geometric Bottlenecks in Single-Stage Training: Observations from the Optimization Manifold

## Abstract

Bi-level meta-learning methods for LLM domain adaptation jointly optimize cross-task generalization and task-specific specialization, coupling these objectives into a single nested optimization. We hypothesize that this coupling induces **gradient interference** under heterogeneous task distributions, forcing models into "compromise" solutions that fail to specialize. To test this hypothesis, we design controlled experiments comparing coupled optimization (MAML-en-LLM) against **TEMPO-LLM**, a temporally staged alternative that separates consolidation, alignment and refinement into sequential stages. Our analysis reveals striking behavioral differences: (1) coupled optimization produces uniformly high gradient similarity across diverse tasks, while temporal staging preserves task-specific directions with substantially higher variance; (2) staged optimization generates significantly sparser adaptation parameters with distinct per-domain "signatures," versus overlapping "barcode-like" patterns. These findings demonstrate that **temporal organization of learning pressures** is a structural degree of freedom in neural network optimization that fundamentally shapes adaptation capacity.

## 1 Introduction

The remarkable empirical success of deep learning has largely outpaced our theoretical understanding of *why* certain algorithmic choices succeed or fail. One persistent mystery concerns the behavior of meta-learning algorithms under task heterogeneity: methods like MAML Finn et al. (2017) and its LLM-specialized variants Sinha et al. (2024); Zhang et al. (2025b) achieve strong transfer on homogeneous task families but exhibit systematic failures when task distributions become diverse Yu et al. (2020); Wallingford et al. (2022). What structural property of these algorithms produces this brittleness?

We investigate a specific hypothesis about the source of this failure. Bi-level meta-learning formulates adaptation as nested optimization, where an outer loop learns cross-task generalization while an inner loop performs task-specific adaptation. Critically, both objectives operate on *shared parameters simultaneously*. Under heterogeneous task distributions, we hypothesize that this coupling forces the optimization into a fundamentally conflicted state: parameters must simultaneously remain stable across tasks (for generalization) while being highly responsive to task-specific gradients (for adaptation). When task optima diverge, the resulting **gradient interference** should trap the model in "compromise" solutions - a barycenter that minimizes average loss but fails to capture the distinct geometric structure required for any single domain.

> **Central Research Question:** Does coupled bi-level optimization produce undifferentiated parameter updates across diverse tasks and does temporally staging learning objectives eliminate this interference to enable genuine task-specific adaptation?

To test this hypothesis, we design controlled experiments using LLM domain adaptation as a testbed. We compare a coupled baseline (MAML-en-LLM) against TEMPO-LLM (**TE**mporal **M**ulti-stage **P**arameter **O**ptimization for **LLM**s), which algorithmically separates the optimization loop into three temporally independent stages: (1) foundation consolidation, (2) knowledge-task alignment and (3) task-specific refinement. Each stage operates on distinct optimization manifolds without seeing

Table 1: **Sci4DL Workshop Themes Alignment Map**

| Workshop Theme | Scientific Contribution | Evidence in Paper | Sec. |
|---|---|---|---|
| **Validate/falsify hypotheses about deep networks** | **Hypothesis-driven investigation**: We formulate and test explicit hypotheses about gradient interference in coupled vs. staged optimization. | Gradient similarity analysis confirms coupled optimization produces undifferentiated updates (Table 3). | §3 |
| **Report observations to inform theoretical models** | **Quantitative behavioral signatures**: We document novel empirical regularities distinguishing optimization regimes. | $6.5\times$ sparsity difference; distinct "signature" vs. "barcode" patterns across adaptation methods (Figures 3, 4). | §3 |
| **Evidence new phenomena or empirical regularities** | **Temporal organization as structural DoF**: We demonstrate that *when* learning pressures act fundamentally shapes adaptation capacity. | Stage ablations reveal alignment is critical - consolidation alone degrades OOD performance (Table 2). | §3 |
| **Develop controlled experimental settings** | **Isolating the coupling variable**: (coupled vs. staged) with matched architecture, data budget and evaluation protocol. | Controlled comparison isolates temporal staging as the sole intervention (Figure 1). | §2 |

gradients from other stages. This temporal staging is the *sole experimental intervention* - all other variables (model architecture, LoRA configuration, data budget) are held constant.

**Contributions.** This paper presents:

- **Gradient conflict analysis** showing that coupled optimization produces uniformly similar gradients while temporal staging preserves task-specific directions with substantially higher variance.
- **Weight sparsity evidence** demonstrating significantly sparser adaptation in temporally staged optimization, with distinct domain-specific "signatures" versus overlapping "barcode" patterns.
- **Stage ablations** revealing that alignment is critical for out-of-domain transfer - consolidation alone can substantially degrade OOD performance (detailed results in Appendix C).

These findings suggest that the *when* of optimization - temporal organization of learning pressures - is an underexplored structural degree of freedom with significant implications for understanding and improving neural network adaptation. Table 1 provides a detailed mapping of our contributions to the workshop's scientific methodology themes.

## 2 EXPERIMENTAL DESIGN

**Problem Setting and Variables.** Let $\mathcal{X}$ and $\mathcal{Y}$ denote input and output spaces. A task $\mathcal{T}$ is defined by a data distribution $p_{\mathcal{T}}(x, y)$ and a loss function $\ell : \mathcal{Y} \times \mathcal{Y} \to \mathbb{R}_+$. We assume access to a distribution of training tasks $p(\mathcal{T})$. We consider a pretrained LLM with frozen parameters $\theta_0$. Adaptation is performed using low-rank adapters (LoRA), parameterized as $W = W_0 + BA^\top$, $A \in \mathbb{R}^{r \times d_{\text{in}}}$, $B \in \mathbb{R}^{d_{\text{out}} \times r}$, where $r \ll \min(d_{\text{in}}, d_{\text{out}})$. All adapter parameters are collected into a vector $\theta^a \in \mathbb{R}^D$. Given a previously unseen task $\mathcal{T}_*$, the objective is to obtain task-adapted parameters $\theta^a_{\mathcal{T}_*}$ using information from $p(\mathcal{T})$ and task-specific context.

**Coupled Baseline.** MAML-en-LLM Sinha et al. (2024) formulates adaptation via nested optimization: $\theta^* = \arg\min_\theta \mathbb{E}_{\mathcal{T} \sim p(\mathcal{T})} \left[ \mathcal{L}_{\mathcal{T}}(\theta'_{\mathcal{T}}) \right]$, where $\theta'_{\mathcal{T}} = \theta - \eta \nabla_\theta \mathcal{L}_{\mathcal{T}}(\theta)$. This formulation *couples* generalization (outer loop) and adaptation (inner loop) on shared parameters $\theta$.

**Staged Condition.** TEMPO-LLM algorithmically separates optimization into three temporally sequenced stages, each operating independently without cross-stage gradient flow:

1. **Stage 1 (Consolidation):** A hyper-convolutional parameter generator learns to map task context to LoRA parameters, trained on supervised regression against task-specific checkpoints. No target-specific gradients. (Details in Appendix E)

2. **Stage 2 (Alignment):** A conditional VAE explicitly aligns consolidated parameters with target task semantics via latent space projection. (Details in Appendix F)

3. **Stage 3 (Refinement):** Lightweight inference-time specialization through bounded RL-based strategy selection without prolonged fine-tuning. (Details in Appendix G)

Crucially, each stage is optimized and applied in strict temporal sequence. This structural intervention directly tests whether *when* learning pressures act - rather than how objectives are weighted - determines adaptation quality.

**Training tasks (6):** Foundation consolidation (Stage 1) is done on the meta-training tasks which are ARC-Challenge, ARC-Easy, HellaSwag, BoolQ, PIQA, WinoGrande (commonsense reasoning). For Out-of-domain evaluation, we use GSM-8K, MATH, DivLogicEval, SocialIQA, CodeMMLU and JAMA Clinical Challenge datasets. This setup ensures substantial semantic distance between training and evaluation domains to stress-test generalization.

## 3 CONTROLLED EXPERIMENTS

### 3.1 EXPERIMENT 1: GRADIENT CONFLICT ANALYSIS

> **Hypothesis 1:** Coupled optimization produces homogeneous gradient directions across diverse tasks; temporal staging preserves task-specific gradient structure.

**Method.** For each condition, we load trained parameters and compute task-specific gradient vectors $\mathbf{g}_\mathcal{T}$ on samples from each OOD task. We report pairwise cosine similarity statistics across all 15 task pairs.

**Results.** Table 3 and Figure 2 reveal a striking difference. MAML-en-LLM produces *uniformly high* gradient similarity (mean 0.884, std 0.047) - gradients from mathematics, medical and coding tasks point in nearly identical directions. This confirms our hypothesis: coupled optimization forces the initialization into an undifferentiated state where task-specific structure is lost.

In contrast, temporal decoupling produces lower mean similarity (0.749) with $2.5\times$ higher variance (0.116). The wider similarity range (0.476–0.940) appropriately reflects semantic relationships: mathematics tasks cluster (GSM–MATH: 0.94) while distant pairs properly diverge (SocialIQA–JAMA: 0.46).

### 3.2 EXPERIMENT 2: LoRA WEIGHT SPARSITY ANALYSIS

> **Hypothesis 2:** Temporal staging enables sparse, focused parameter modifications; coupling produces diffuse, distributed updates.

**Method.** We analyze LoRA weight matrices across all 6 OOD tasks, computing sparsity (fraction of near-zero weights), L1/L2 norms and maximum weight magnitudes.

**Results.** Table 4 confirms Hypothesis 2. Temporal decoupling produces **6.5$\times$ higher sparsity** (10.0% vs. 1.5%), indicating that adaptation concentrates on fewer parameters. Simultaneously, overall norms are substantially lower (65% reduction in L1) while *peak magnitudes are higher* ($1.8\times$). This pattern - fewer parameters carrying the adaptation signal with higher individual impact - is characteristic of focused, task-specific updates rather than diffuse parameter spreading forced by gradient averaging.

### 3.3 EXPERIMENT 3: VISUAL ADAPTATION SIGNATURES

> **Hypothesis 3:** Coupled optimization produces overlapping "barcode-like" patterns; staging produces distinct "signatures" for each domain.

**Method.** We extract LoRA matrices from a representative layer (Layer 12) and compute the effective weight update $\Delta W = \frac{\alpha}{r} BA$. We visualize the first 200 parameters as line plots (each task as a colored line) and as heatmaps (tasks as rows).

**Results.** The visual evidence is unambiguous. Figure 3 shows that MAML-en-LLM produces *overlapping lines* - the model learns identical parameters regardless of whether adapting to medical, coding, or mathematical domains. This "barcode" pattern directly visualizes the compromise solution forced by gradient interference.

Temporal decoupling produces qualitatively different behavior: the lines clearly separate, with distinct "signatures" for each domain visible in both the line plot (Figure 3) and heatmap (Figure 4). This

visual proof complements our quantitative findings and provides an intuitive understanding of why temporal separation enables genuine specialization.

### 3.4 EXPERIMENT 4: STAGE ABLATION

> **Hypothesis 4:** Each temporal stage serves a distinct function; consolidation without alignment can harm OOD transfer.

**Method.** We evaluate three conditions: (1) Base model (no adaptation), (2) Stage 1 only (consolidation without alignment/refinement), (3) Full pipeline (all three stages).

**Results.** Figure 1 and table 2 reveals a critical finding: Stage 1 alone (consolidation) can *significantly degrade* OOD performance. On distant domains like mathematics, consolidating knowledge from commonsense tasks *hurts* performance substantially. The full pipeline recovers this performance, demonstrating that alignment (Stage 2) is essential for semantic compatibility before specialization occurs (see Appendix for detailed per-task breakdowns).

This ablation directly tests whether temporal staging is merely "more stages" or whether each stage serves a functionally distinct role. The answer is clear: consolidation without alignment and refinement produces *negative transfer*; temporal organization matters not just for preventing interference but for enabling cross-domain compatibility.

**Performance Evidence** The behavioral differences documented above correspond to substantial performance gains (Table 2). On in-domain tasks, TEMPO-LLM consistently improves over MAML-en-LLM. On out-of-domain tasks - the critical test of adaptive specialization - gains are particularly pronounced on distant domains like mathematics. These results confirm that the mechanistic differences (gradient diversity, sparsity, distinct signatures) translate to measurable capability improvements.

## 4 DISCUSSION AND IMPLICATIONS

**Summary.** Our controlled experiments provide consistent evidence that coupled bi-level optimization produces gradient interference under task heterogeneity. MAML-en-LLM exhibits uniformly high gradient similarity, low-sparsity diffuse updates and identical "barcode" patterns across tasks. TEMPO-LLM - via temporal staging alone - produces qualitatively different behavior: task-appropriate gradient clustering, significantly sparser focused updates and distinct domain-specific signatures.

**Implications.** These findings suggest that **temporal organization of learning pressures** is an underexplored structural degree of freedom in neural network optimization. The "when" of optimization fundamentally shapes what representations the network can learn, paralleling cognitive learning theory where mastery of prerequisites precedes transfer to new domains Bloom (1968).

**Limitations.** Experiments use Qwen2.5 models (0.5B/1.5B); validation on larger models would strengthen generality. Behavioral analyses focus on LoRA; full-model fine-tuning and multi-turn settings remain future work.

## 5 CHALLENGE: SCIENCE OF DL IMPROVEMENT

> **Submission for the Challenge: Science of Deep Learning Improvement**

**1. What model are you targeting?** We target **Large Language Models (LLMs) undergoing domain adaptation** via meta-learning. Specifically, we address the failure modes of bi-level optimization algorithms (like MAML and its LLM variants) when applied to heterogeneous task distributions. The current literature relies heavily on coupled optimization, where cross-task generalization and task-specific specialization are optimized simultaneously on shared parameters Sinha et al. (2024); Zhang et al. (2025b). While effective for homogeneous tasks, this coupling induces catastrophic gradient interference when tasks diverge semantically, trapping models in "compromise" solutions that fail to specialize efficiently Yu et al. (2020); Wallingford et al. (2022). Our work targets this fundamental structural limitation by proposing TEMPO-LLM, a model of adaptation that temporally decouples consolidation, alignment, and refinement.

**2. How do your results contribute to understanding these models?** Our results provide concrete empirical evidence for **gradient interference** as the mechanistic cause of adaptation failure in coupled systems.

- **Gradient Geometry:** We show that coupled optimization produces universally high gradient cosine similarity ($\mu = 0.88$) across distinct domains (Math, Code, Medicine), effectively erasing task-specific geometric structure. In contrast, temporal staging restores natural gradient diversity ($\mu = 0.75$, $2.5\times$ higher variance).
- **Sparsity & Localization:** We uncover a direct link between optimization structure and parameter sparsity. Coupled updates result in diffuse, dense weight changes ("barcode" patterns, 1.5% sparsity), whereas staged optimization yields highly sparse (10%), magnitude-rich updates with distinct per-task signatures.
- **Role of Alignment:** Our ablation studies reveal that knowledge consolidation alone is insufficient and can be harmful (negative transfer) without an explicit alignment phase, clarifying the functional necessity of intermediate representations in transfer learning.

**3. How do you expect your submission to influence future work?** We propose that the **temporal organization of learning pressures** is a critical, under-utilized structural degree of freedom in neural network design.

- **Design Principle:** Future work should move beyond simply weighting competing objectives (regularization) to temporally sequencing them. This suggests a shift from "what to optimize" to "when to optimize" different capabilities.
- **Modular Adaptation:** Our findings encourage the development of modular adaptation architectures where consolidation (generalization) and refinement (specialization) are treated as distinct, sequentially optimized phases rather than a single joint problem.
- **Scientific Methodology:** By demonstrating how specific optimization choices (coupling vs. staging) dictate the geometric properties of the solution (sparsity, gradient direction), we provide a template for diagnosing optimization bottlenecks in other large-scale learning systems.

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

## APPENDIX NAVIGATION INDEX

# A MOTIVATION

## A.1 EMPIRICAL FAILURE MODES OF CURRENT APPROACHES

Current approaches to LLM domain adaptation exhibit recurring failure modes under task heterogeneity. Bi-level meta-learning jointly optimizes cross-task generalization and task-specific adaptation using shared parameters, forcing them to remain stable across tasks while simultaneously responding to task-specific gradients Yu et al. (2020); Qin et al. (2023). When task optima diverge, this coupling induces parameter interference and unstable adaptation, a phenomenon consistently observed under heterogeneous task distributions Wallingford et al. (2022); Jeong & Yoon (2025). Self-play and self-evolution methods remove explicit coupling by relying on self-generated curricula and autonomous refinement, but inherit a complementary limitation: effective practice requires tasks calibrated to current competence and informative feedback, conditions that cannot be satisfied when curriculum quality is endogenously bounded by the learner's existing knowledge (Ericsson et al., 1993; Zhao et al., 2024). In specialized or out-of-distribution domains, this leads to uninformative supervision and negative transfer rather than progressive improvement Briesch et al. (2023); Wang et al. (2026).

These failures are structural rather than incidental. In bi-level optimization, the conflict between stability and plasticity follows directly from the nested objective formulation: increasing task-specific adaptability necessarily perturbs representations intended for cross-task generalization and no amount of regularization or capacity scaling eliminates this trade-off without altering the optimization structure itself Ding et al. (2022); Chen et al. (2023); Zhang et al. (2024). Likewise, self-play methods cannot be repaired by scaling synthetic data generation, because the quality of generated supervision is constrained by the learner's pre-aligned knowledge. More fundamentally, both paradigms assume that knowledge consolidated from source tasks can be directly reused in new semantic regimes-contradicting transfer-learning evidence that successful reuse requires an intermediate alignment phase before specialization (Singley & Anderson, 1989). Analogous phenomena have been observed in modern neural systems, where task composition succeeds only when tasks are aligned or non-

interfering and breaks down under contradictory regimes due to parameter interference and gradient sign disagreement (Li et al., 2025).

The key insight motivating this work is that **consolidation, alignment and specialization impose incompatible requirements when optimized jointly**. Enforcing temporal separation between these roles avoids parameter interference by construction and prevents negative transfer arising from semantic mismatch.

## B  RELATED WORK

Meta-learning enables rapid task adaptation by leveraging shared structure across task distributions (Schmidhuber, 1987; Thrun & Pratt, 1998; Hospedales et al., 2021). Classical approaches like Matching Networks (Vinyals et al., 2016), Prototypical Networks (Snell et al., 2017) and MAML (Finn et al., 2017) established bi-level optimization paradigms (Vilalta & Drissi, 2002; Nichol et al., 2018), but their application to LLMs faces unique scalability challenges.

### B.1  GRADIENT-BASED META-LEARNING FOR LLMS

The Model-Agnostic Meta-Learning (MAML) family exemplifies gradient-based approaches that optimize for initial parameters from which tasks can be adapted with a few gradient steps. These methods have been effective on standard few-shot classification and reinforcement learning benchmarks. However, they inherently bind generalization and adaptation through a shared optimization trajectory, requiring second-order gradients or multiple inner-loop updates that are costly for large models. Variants such as MAML-en-LLM Sinha et al. (2024), MLtD Hou et al. (2022), ReptiLoRA Kim et al. (2025) adapt this framework to large pretrained language models, demonstrating improved unseen domain and adaptation performance via meta-training tailored for LLMs. Although promising, these approaches still adhere to coupled adaptation dynamics and necessitate gradient backpropagation through the LLM's entire parameter space, which remains expensive and often impractical at scale.

### B.2  BAYESIAN AND AMORTIZED META-LEARNING FOR LLMS

Probabilistic meta-learning models introduce a generative perspective on task parameters, learning priors or distributions that can produce task-specific models. Hierarchical Bayesian frameworks such as LiFT Kim & Hospedales (2025)(Learning to Fine-Tune) treat PEFT adapter parameters of LLMs as random variables governed by a higher-level latent prior that captures shared information across tasks. By performing efficient sampling and inference, such models offer better generalization and can outperform traditional meta-learning and heuristic mixing approaches. More recently, Amortized Bayesian Meta-Learning for LoRA (ABMLL) Zhang et al. (2025a) has been proposed specifically for large language models. ABMLL adapts amortized Bayesian meta-learning techniques to the LoRA parameterization of LLMs, reframing global and task-specific parameters to improve computational efficiency and generalization on multi-task benchmarks such as Unified-QA and CrossFit. The Bayesian formulation also yields improved uncertainty quantification for LLM adaptation. While these Bayesian approaches enable principled incorporation of uncertainty and task parameter distributions, they still rely on gradient-based inference or sampling within the meta-training loop, thereby retaining some of the limitations of coupled meta-learning, particularly for large pretrained models.

### B.3  ADAPTATION BEYOND GRADIENTS

While the above methods primarily rely on gradients or approximate inference within the meta-training paradigm, emerging research suggests alternative adaptation strategies for large models that do not depend on gradient descent. For instance, in-context meta-learning Coda-Forno et al. (2023); Chen et al. (2022) explores recursive in-context learning capabilities of LLMs to improve task performance without parameter updates.

Our framework diverges from these approaches by explicitly decoupling the generalization and adaptation processes, learning a generative model over task parameters in a task-agnostic manner and adopting closed-loop reinforcement learning for adaptation that does not require gradients through the meta-learner or LLM parameters. This enables modular adaptation policies and addresses scalability and flexibility limitations inherent in prior gradient-coupled meta-learning approaches.

## C    EMPIRICAL VALIDATION

We compare TEMPO-LLM against the following adaptation strategies: (i) *No Meta-Train LoRA*, which performs task-specific LoRA fine-tuning without meta-training; (ii) *Union Train LoRA*, which trains a single LoRA adapter on the union of all training tasks; (iii) *MAML-en-LLM* (Sinha et al., 2024); (iv) *ABMLL* (Zhang et al., 2025b).

| In-Domain Tasks | | | | | | | |
|---|---|---|---|---|---|---|---|
| **Dataset** | ARC-c | ARC-e | HellaSwag | BoolQ | PIQA | WinoGrande | **Avg** |
| *Qwen2.5-1.5B-Instruct* | | | | | | | |
| Base Model | 71.5 | 83.0 | 50.9 | 56.3 | 45.8 | 50.6 | 59.6 |
| No Meta-Train LoRA | **74.5** | 84.4 | 55.8 | 55.6 | 65.6 | 48.2 | 64.0 |
| Union Train LoRA | 63.2 | 73.9 | 48.9 | 55.1 | 47.8 | **61.3** | 58.3 |
| ABMLL | 69.9 | 83.2 | 51.1 | **63.2** | 54.3 | 52.9 | 62.4 |
| MAML-en-LLM | 66.0 | 84.3 | **59.3** | 58.7 | 68.1 | 56.8 | 65.5 |
| TEMPO-LLM (Stage 1) | 73.0 | 83.7 | 56.2 | 55.2 | 56.4 | 50.2 | 62.5 |
| **TEMPO-LLM (Full)** | 73.7 | **88.4** | 57.2 | 58.8 | **70.7** | 57.3 | **67.7** |
| *Qwen2.5-0.5B-Instruct* | | | | | | | |
| Base Model | 38.3 | 54.8 | 26.5 | 37.0 | 16.6 | 50.2 | 37.2 |
| No Meta-Train LoRA | 40.7 | 59.4 | 23.4 | 22.1 | 66.2 | 35.7 | 41.2 |
| Union Train LoRA | 39.7 | 47.4 | 26.3 | 14.7 | 51.1 | 50.5 | 38.3 |
| ABMLL | 37.6 | 54.4 | 26.5 | **62.2** | 37.6 | 34.5 | 42.1 |
| MAML-en-LLM | 47.7 | 63.7 | 36.3 | 46.2 | 67.7 | 50.1 | 51.9 |
| TEMPO-LLM (Stage 1) | 42.7 | 63.2 | 25.9 | 44.9 | 47.6 | 50.0 | 45.7 |
| **TEMPO-LLM (Full)** | **55.5** | **74.7** | **48.3** | 58.7 | 60.1 | **52.8** | **58.4** |

| Out-of-Domain Tasks | | | | | | | |
|---|---|---|---|---|---|---|---|
| **Dataset** | GSM-8K | MATH | DivLogicEval | SocialIQA | CodeMMLU | JAMA | **Avg** |
| *Qwen2.5-1.5B-Instruct* | | | | | | | |
| Base Model | 51.4 | 30.3 | 28.3 | 65.9 | 42.6 | 38.9 | 42.8 |
| Union Train LoRA | 34.2 | 32.2 | 24.1 | 51.4 | 34.7 | 34.7 | 36.1 |
| ABMLL | 28.7 | 15.9 | 26.9 | 66.3 | 39.6 | 28.5 | 34.3 |
| MAML-en-LLM | 35.6 | 43.5 | 31.2 | 68.7 | 42.3 | 32.5 | 42.3 |
| TEMPO-LLM (Stage 1) | 32.6 | 40.1 | 28.6 | 68.6 | 44.1 | 39.5 | 42.2 |
| **TEMPO-LLM (Full)** | **51.8** | **46.9** | **31.4** | **69.5** | **44.6** | **41.5** | **47.6** |
| *Qwen2.5-0.5B-Instruct* | | | | | | | |
| Base Model | 15.2 | 2.8 | 22.4 | 50.8 | 32.4 | 23.8 | 24.5 |
| Union Train LoRA | 15.6 | 6.8 | 20.3 | 39.5 | 29.8 | 29.9 | 29.9 |
| ABMLL | 20.4 | 7.1 | 23.7 | 53.1 | 28.2 | 16.8 | 24.9 |
| MAML-en-LLM | 29.1 | **26.3** | 25.1 | 54.9 | 34.1 | 26.4 | 32.6 |
| TEMPO-LLM (Stage 1) | 20.8 | 24.1 | 21.0 | 33.5 | 29.1 | 11.7 | 25.7 |
| **TEMPO-LLM (Full)** | **30.3** | 24.5 | **28.7** | **55.1** | **35.6** | **31.2** | **34.2** |

Table 2: Performance comparing **TEMPO-LLM** (split into Stage 1 consolidation and full adaptation) against benchmarks. **Bold**: best performance.

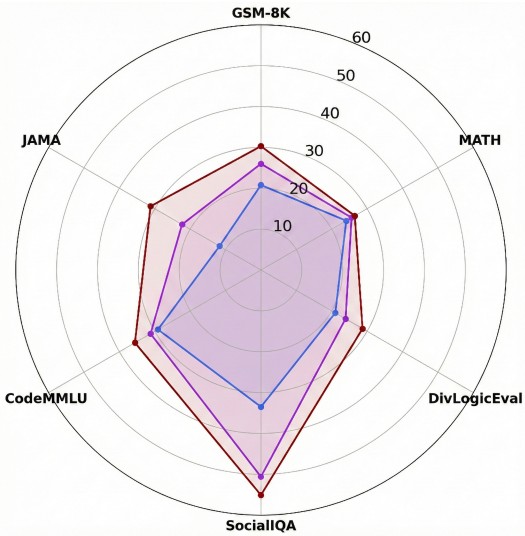

Figure 1: Ablation Study revealing the incremental effectiveness of each TEMPO-LLM stage with Stage I, Stage II and Stage III

# D   VISUALS

Table 3: **Gradient cosine similarity across 6 OOD tasks.** Lower mean with higher variance indicates better task-specific differentiation.

| Condition | Mean | Std | Min | Max |
|---|---|---|---|---|
| Coupled (MAML) | 0.884 | 0.047 | 0.770 | 0.976 |
| **TEMPO-LLM** | **0.749** | **0.116** | 0.476 | 0.940 |

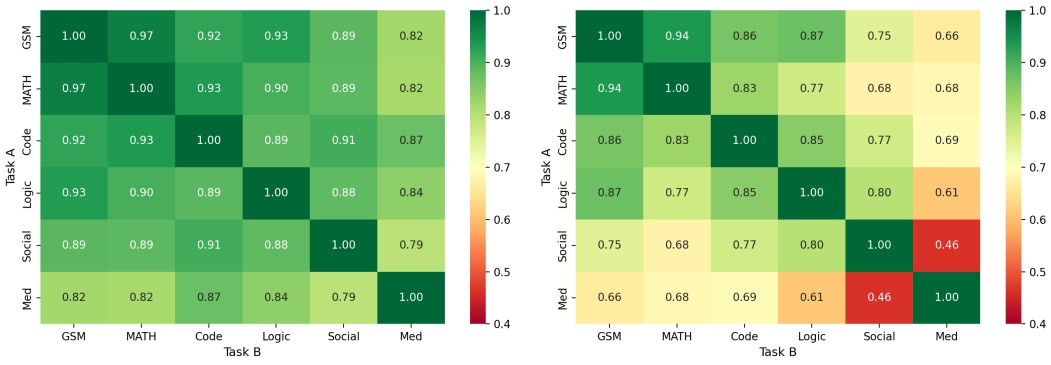

Figure 2: **Pairwise gradient similarity.** Coupled optimization (left) shows uniformly high similarity across all task pairs. TEMPO-LLM (right) shows appropriate clustering - math tasks (GSM–MATH: 0.94) group tightly while semantically distant pairs diverge (Social–Medical: 0.46).

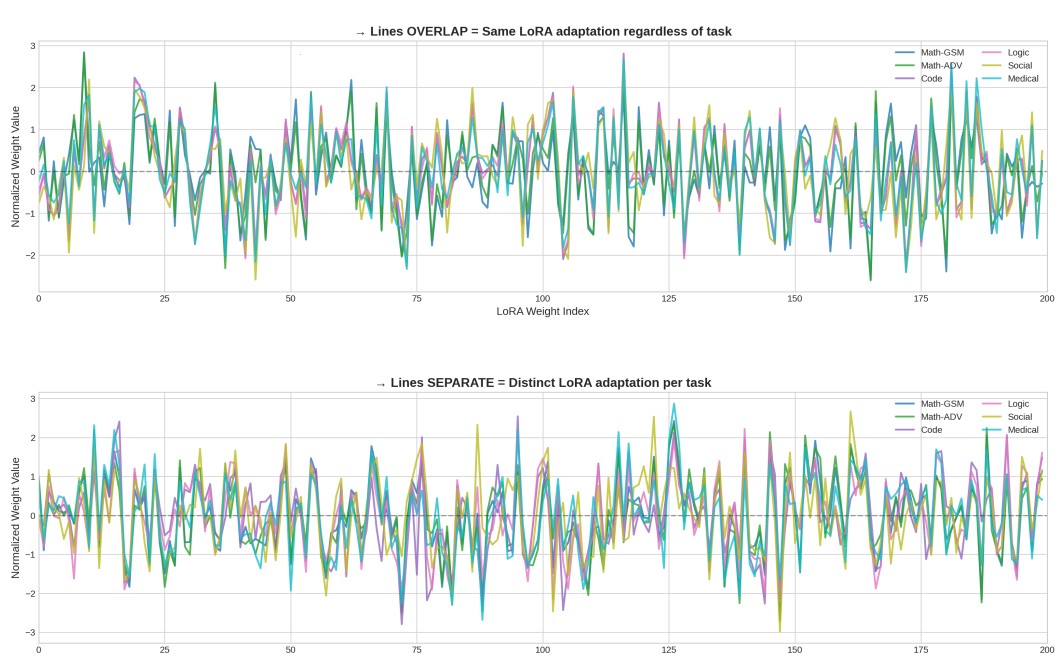

Figure 3: **LoRA weight pattern visualization.** Coupled (top): lines overlap completely - identical weights for all tasks. TEMPO-LLM (bottom): distinct curves for Code, Medical, Math domains.

Table 4: **LoRA weight statistics across 6 OOD tasks.**

| Metric | Coupled | TEMPO-LLM | Ratio |
|--------|---------|-----------|-------|
| Weight Sparsity | 1.55% | **10.01%** | 6.5× higher |
| L2 Norm | 0.575 | **0.400** | 30% lower |
| L1 Norm | 2512 | **866** | 65% lower |
| Max Weight | 0.0017 | **0.0031** | 1.8× higher |

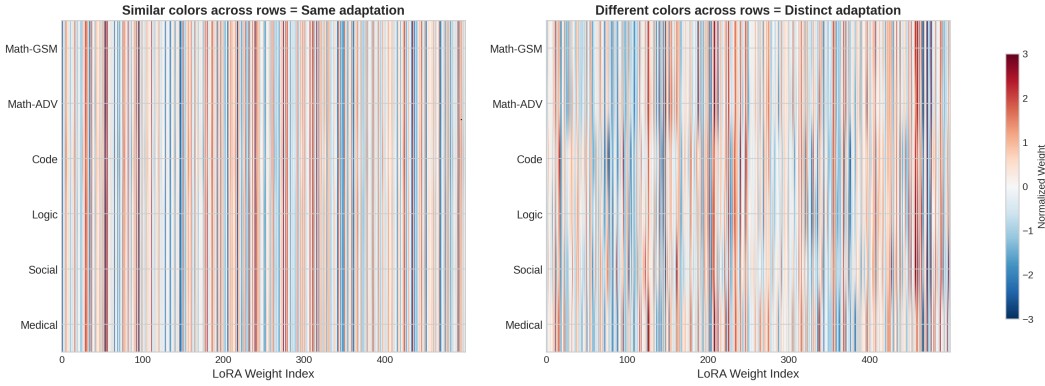

Figure 4: **LoRA weight heatmaps.** Coupled (left): uniform vertical stripes confirm identical parameters across tasks. TEMPO-LLM (right): varied patterns demonstrate task-specific adaptations.

| Task | Pretraining | | | | Fine-Tuning | |
|---|---|---|---|---|---|---|
| | Learning Rate | Steps | Batch Size | # Samples | Learning Rate | Steps |
| Common Sense | $1 \times 10^{-4}$ | 75 | 32 | 5,000 | $1 \times 10^{-5}$ | 50 |

Table 5: Checkpoint collection settings for LoRA parameter generation

# E  STAGE 1 : FOUNDATIONAL KNOWLEDGE CONSOLIDATION MODULE

This section provides additional implementation details for the foundation consolidation stage (Stage 1) of TEMPO-LLM, focusing on (i) LoRA checkpoint collection and (ii) hyper-convolutional decoder architectures.

## E.1  PROMPT ENCODING TENSOR STRUCTURE

In Eq. (4), prompt embeddings are constructed as

$$
\begin{aligned}
\mathbf{c}_i &= \text{Encoder}(p_i; \theta_{\text{enc}}), \\
\mathbf{C} &= [\mathbf{c}_1; \ldots; \mathbf{c}_N] \in \mathbb{R}^{B \times N \times L \times C}
\end{aligned}
\tag{1}
$$

Here:

- $B$ denotes the batch size (number of tasks processed in parallel),
- $N$ is the number of representative prompts sampled per task,
- $L$ is the token sequence length of each prompt,
- $C$ is the hidden embedding dimension of the frozen text encoder.

This tensorized representation follows the prompt-to-weights formulation introduced in Drag-and-Drop LLMs (DnD) (Liang et al., 2025), and enables convolutional processing over task, prompt and token dimensions. Since $B$ does not affect architectural design, it is omitted in subsequent architecture descriptions.

## E.2  LoRA CHECKPOINT COLLECTION PROTOCOL

TEMPO-LLM relies on a supervised prompt-to-weights mapping trained using LoRA checkpoints collected from task-specific fine-tuning runs. We adopt the checkpoint collection procedure proposed in DnD (Liang et al., 2025).

Each checkpoint collection run consists of two phases:

- **Pretraining phase**: the base model is trained on the target dataset for a fixed number of steps using a higher learning rate.
- **Fine-tuning phase**: training continues with a lower learning rate, and a LoRA checkpoint is saved at each step.

Except for learning rate and training steps, all other hyperparameters (e.g., batch size, optimizer, data sampling strategy) are kept identical between the two phases. For datasets with fewer samples than the specified number, the full dataset is used.

Table 5 summarizes the checkpoint collection settings for different task families.

## E.3  PARAMETER TOKENIZATION

We employ a parameter tokenization strategy following Wang et al. (2025), which transforms LoRA adapter weights into a sequence of uniform tokens suitable for processing by the generalization model. It involves,

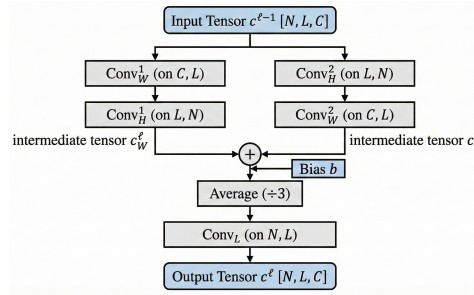

Figure 5: (a) Tokenization of LoRA parameters *(left)* and (b) Foundational Knowledge Consolidation Module Architecture *(right)*

| Size | Channel Progression $(N, L, C)$ |
|------|--------------------------------|
| 0.5B | $(n, 384, 384) \rightarrow (n, 200, 300) \rightarrow (n, 100, 256)$ |
|      | $(2n, 50, 200) \rightarrow (4n, 50, 200) \rightarrow (8n, 25, 200)$ |
|      | $(8n, 10, 200) \rightarrow (16n, 10, 200) \rightarrow (4296, 8, 128)$ |
| 1.5B | $(n, 384, 384) \rightarrow (n, 200, 300) \rightarrow (n, 100, 256)$ |
|      | $(2n, 50, 200) \rightarrow (4n, 50, 200) \rightarrow (8n, 25, 200)$ |
|      | $(8n, 10, 200) \rightarrow (16n, 10, 200) \rightarrow (4508, 18, 258)$ |

Table 6: Hyper-convolutional decoder architectures used for LoRA parameter generation across model sizes ($n < 128$)

**Layer-wise splitting & normalization-** Given complete LoRA adapter parameters $W$ spanning all layers, first parameters are segregated by layer index and then layer-wise normalization is applied to reduce distribution shifts across layers:

$$W \xrightarrow{\text{split by layer}} [w[1], \dots, w[I]]$$
$$\xrightarrow{\text{normalize}} [\hat{w}[1], \dots, \hat{w}[I]] \tag{2}$$

**Uniform tokenization-** Each normalized layer $\hat{w}[i]$ is then partitioned into contiguous, non-overlapping chunks of uniform size $k$ (with padding applied to the final chunk if necessary):

$$\hat{w}[i] \xrightarrow{\text{tokenize}} K[i] = \left[ k_i^1, k_i^2, \dots, \text{pad}\left(k_i^{J_i}\right) \right], \tag{3}$$

where $J_i$ denotes the number of tokens for layer $i$ and pad($\cdot$) indicates zero-padding to achieve uniform token length $k$. Each checkpoint $W$ is then assigned a unique permutation state $S$ encoded as a one-hot vector. Each token is further augmented with 2D sinusoidal position embeddings. For the $j$-th token in layer $i$, $e_i^j = \text{PE}_{2D}(i, j)$, is computed, where the first dimension encodes layer index $i$ and the second dimension encodes in-layer token position $j$.

For Qwen2.5-0.5B-Instruct with LoRA rank $r = 8$, each layer's LoRA matrices have dimensions $8 \times 896$. With token size $k = 1024$, we obtain 7 tokens of size $8 \times 128$ per layer, with the final token padded to $10 \times 130$. For Qwen2.5-1.5B-Instruct with $r = 16$, matrices of size $16 \times 1536$ decompose into 6 tokens of $16 \times 256$, padded to $18 \times 258$. These tokenization schemes balance information density with computational tractability.

### E.4 HYPER-CONVOLUTIONAL DECODER ARCHITECTURES

The hyper-convolutional decoder maps prompt embedding tensors $\mathbf{C} \in \mathbb{R}^{N \times L \times C}$ to tokenized LoRA parameters. We denote decoder architectures using a tuple $(N, L, C)$ representing the input prompt dimension, token length and channel width respectively. Table 6 lists the decoder architectures used for different foundation model sizes.

### E.5  Hardware and Compute.

All experiments were conducted on a single HPC node running Ubuntu 22.04.1. The system was equipped with an AMD EPYC 8434P CPU (48 physical cores, 96 logical threads), 256 GB of system RAM and four NVIDIA RTX A6000 GPUs, each with 48 GB of dedicated VRAM. GPU-accelerated workloads were executed using CUDA 12.4 and all experiments were implemented in Python 3.12.11.

The same hardware configuration was used for all methods, including our approach and all baselines, ensuring identical compute conditions and avoiding hardware-induced advantages. No system-level optimizations or specialized infrastructure beyond standard single-node multi-GPU execution were employed; reported performance differences therefore reflect algorithmic effects rather than differences in computational resources.

### E.6  Hyperparameter Summary

This section summarizes hyperparameters and configuration choices that are explicitly specified in the main paper and appendix, collected here for ease of reproducibility.

**LoRA Configuration.**  All experiments use LoRA adapters with identical parameterization across methods. For Qwen2.5-0.5B-Instruct, the LoRA rank is $r = 8$, yielding adapter matrices of size $8 \times 896$. For Qwen2.5-1.5B-Instruct, the rank is $r = 16$, yielding matrices of size $16 \times 1536$. LoRA adapters are applied to the query, key, value and output projections in attention layers, as well as the gate, up and down projections in MLP blocks. All base model parameters remain frozen.

**Foundation Consolidation (Stage 1).**  LoRA checkpoints used for training the hyper-convolutional parameter generator are collected using a two-phase procedure. In the pretraining phase, models are trained for 75 steps with learning rate $1 \times 10^{-4}$. In the fine-tuning phase, training continues for 50 steps with learning rate $1 \times 10^{-5}$. Both phases use a batch size of 32 and up to 5,000 samples per dataset, as summarized in Table 5.

**Prompt Encoding and Decoder Architecture.**  Task prompts are encoded using a frozen Sentence-BERT (all-MiniLM-L6-v2) encoder with maximum sequence length 384. The hyper-convolutional decoder consists of cascaded convolutional blocks, each containing five convolutional layers, with channel progressions specified in Table 6.

### E.7  Dataset Descriptions

We evaluate DECOUPLED on a diverse collection of benchmarks spanning commonsense reasoning, mathematics, logic, social reasoning, medical question answering and code understanding. Following prior meta-learning and parameter-generation work (Liang et al., 2025), we distinguish between *in-domain* tasks used during foundation consolidation and *out-of-domain* tasks used solely for evaluation.

**In-Domain Tasks.**  By in-domain tasks, we refer to tasks that are included during the foundation consolidation stage (Stage 1), following a leave-one-out meta-training protocol. ARC-Challenge and ARC-Easy (Clark et al., 2018) contain grade-school-level multiple-choice science questions designed to test elementary reasoning. HellaSwag (Zellers et al., 2019) evaluates commonsense inference by requiring models to select the most plausible continuation of a given context from adversarially constructed alternatives. BoolQ (Clark et al., 2019) consists of naturally occurring yes/no questions derived from real-world passages. PIQA (Bisk et al., 2019) focuses on physical commonsense reasoning in everyday situations, requiring selection of the most plausible solution. WinoGrande (Sakaguchi et al., 2021) is a large-scale dataset for commonsense reasoning framed as a fill-in-the-blank task with binary choices.

**Out-of-Domain Tasks.**  Out-of-domain tasks are never used during foundation consolidation and serve to evaluate robustness under domain shift. GSM-8K (Cobbe et al., 2021) consists of grade-school mathematical word problems requiring multi-step arithmetic reasoning. MATH (Hendrycks et al., 2021) contains challenging competition-level mathematics problems spanning algebra, geometry and number theory. DivLogicEval (Chung et al., 2025) assesses logical reasoning through

counterintuitive natural-language questions designed to isolate pure logical inference. SocialIQA (Sap et al., 2019) evaluates social and emotional commonsense reasoning in everyday interactions. CodeMMLU (Manh et al., 2025) is a multitask benchmark for code understanding, covering program analysis, bug detection and software engineering concepts across multiple programming languages. The JAMA Clinical Challenge (Chen et al., 2025) consists of expert-curated medical case questions with detailed explanations, designed to evaluate clinical reasoning and decision-making.

# F   STAGE 2: KNOWLEDGE-TASK ALIGNMENT – IMPLEMENTATION DETAILS

This section provides detailed implementation information for the Fusion VAE used in Stage 2 (Knowledge-Task Alignment).

## F.1   TRAINING DATA PREPARATION

The LoRA checkpoints collected during Stage 1 for training the parameter generator are reused for Fusion VAE training. For each checkpoint, we compute its corresponding task vector using the training dataset of that task. Specifically:

- LoRA parameters are flattened into 1D vectors for processing
- Task vectors are computed as the difference between fine-tuned and pre-trained model outputs on task-specific prompts
- Training pairs $(l^{(i)}, v_{\mathcal{T}_i})$ are constructed from LoRA parameters and corresponding task vectors

## F.2   FUSION VAE ARCHITECTURE

The VAE employs a 1D convolutional encoder-decoder architecture:

**Encoder.**

- 5-layer 1D CNN with channel progression $[1 \rightarrow 32 \rightarrow 64 \rightarrow 128 \rightarrow 256 \rightarrow 512]$
- Kernel size: 4, stride: 4
- Adaptive average pooling to spatial dimension 32
- Separate linear heads for mean $\mu$ and log-variance $\log \sigma^2$

**Task Vector Integration.**   Task vectors are projected to 512 dimensions via a linear layer and concatenated with compressed LoRA features before latent encoding. The decoder receives both the sampled latent $\mathbf{z}$ and the projected task vector for reconstruction.

**Decoder.**   Mirror architecture of the encoder using transposed convolutions, with final output matching the original LoRA parameter dimension.

## F.3   TRAINING PROCEDURE

Training follows a meta-learning framework with the following hyperparameters:

- **Meta-epochs**: 4000
- **Inner loop steps**: 1
- **Inner learning rate**: $1 \times 10^{-3}$
- **Meta learning rate**: $1 \times 10^{-4}$
- **KL weight**: Annealed from 0 to 0.005 over training
- **Optimizer**: Adam with default parameters

The loss function combines reconstruction error (MSE) and KL divergence regularization:

$$\mathcal{L}_{\text{VAE}} = \mathcal{L}_{\text{recon}} + \lambda_{\text{KL}} \cdot \mathcal{L}_{\text{KL}} \tag{4}$$

## F.4   ALGORITHMIC FORMULATION OF STAGE II ALIGNMENT

For completeness, we provide the explicit optimization procedure used to train the task-aware conditional VAE in Stage II. The objective is to transform the generated adapter parameters from Stage I into semantically aligned parameters that respect the target task geometry before refinement.

---

**Algorithm 1** Stage II: Task-Conditioned Fusion Alignment via Conditional VAE

---

**Require:** Training tasks $\{\mathcal{T}_i\}$, generator outputs $\theta_{\text{gen}}^a$, encoder/decoder parameters $(\phi, \psi)$, KL weight $\lambda_{\text{KL}}$

1: **for** each alignment iteration **do**
2:     Sample task $\mathcal{T}_i \sim p(\mathcal{T})$
3:     Compute task semantic signature $\mathbf{v}_{\mathcal{T}_i}$
4:     Encode conditional posterior:

$$(\mu_i, \sigma_i) \leftarrow q_\phi\big(\mathbf{z} \mid \theta_{\text{gen}}^a, \mathbf{v}_{\mathcal{T}_i}\big)$$

5:     Sample latent:

$$\mathbf{z}_i \sim \mathcal{N}(\mu_i, \text{diag}(\sigma_i^2))$$

6:     Decode aligned adapter:

$$\theta_{\text{aligned}}^a \leftarrow \text{Dec}_\psi(\mathbf{z}_i, \mathbf{v}_{\mathcal{T}_i})$$

7:     Reconstruction loss:

$$\mathcal{L}_{\text{recon}} \leftarrow \big\|\theta_{\text{gen}}^a - \theta_{\text{aligned}}^a\big\|^2$$

8:     KL regularization:

$$\mathcal{L}_{\text{KL}} \leftarrow D_{\text{KL}}(q_\phi(\mathbf{z} \mid \cdot) \,\|\, \mathcal{N}(0, I))$$

9:     Full alignment objective:

$$\mathcal{L}_{\text{align}} \leftarrow \mathcal{L}_{\text{recon}} + \lambda_{\text{KL}} \mathcal{L}_{\text{KL}}$$

10:     Update encoder/decoder:

$$(\phi, \psi) \leftarrow (\phi, \psi) - \eta \nabla_{\phi, \psi} \mathcal{L}_{\text{align}}$$

11: **end for**
12: **Return:** aligned adapter distribution $\theta_{\text{aligned}}^a$ for downstream refinement.

---

## F.5 Inference Pipeline

At inference time for an unseen task:

1. **Initial LoRA Generation**: Generate initial LoRA parameters using the Stage 1 parameter generator from target task prompts

2. **Task Vector Computation**: Compute task vector from target task prompts using the pre-trained model

3. **VAE Encoding**: Pass generated LoRA parameters and task vector through the Fusion VAE encoder to obtain latent representation

4. **Aligned Decoding**: Sample from the latent distribution and decode conditioned on the task vector to obtain aligned LoRA parameters

This process ensures that the output LoRA parameters are semantically aligned with the target task while preserving the foundational knowledge consolidated in Stage 1.

# G  STAGE 3: TASK-SPECIFIC REFINEMENT

This section provides a comprehensive treatment of the four adaptation strategy families available in Stage III of DECOUPLED: Test-Time Learning (TTL), Two-Subspace Mixing LoRA, Test-Time Scaling (TTS) and Latent Space Modification. Each strategy represents a bounded inference-time transformation that can be selected by the reinforcement learning–based refinement policy $\pi_\phi$ to optimize task-specific performance.

We cast refinement as a decision-making problem:

$$\alpha^* = \arg\max_{\alpha \in \mathcal{A}} \mathbb{E}_{\alpha \sim \pi_\phi(\cdot|\tau)}[\, R(\tau, \alpha)\,], \tag{5}$$

where each $\alpha \in \mathcal{A} = \{\texttt{TTL}, \texttt{LoRA}, \texttt{TTS}, \texttt{Latent}, \dots\}$ specifies a concrete adaptation operator with an explicit hyperparameter configuration. The strategy model emits structured JSON outputs that are parsed deterministically at deployment.

## G.1  TEST-TIME LEARNING (TTL)

Test-Time Learning adapts the model using only unlabeled test inputs by minimizing input perplexity. The key insight is that reducing perplexity on a question $x$ implicitly improves answer quality when the question-answer pair is semantically aligned.

**Problem Setting.**  Let $f_\Theta$ be a pretrained autoregressive LLM with frozen base parameters $\Theta$. Given unlabeled test inputs $\mathcal{D}_{\text{Test}} = \{x_j\}_{j=1}^M$, TTL adapts the model by updating only lightweight LoRA parameters $\Delta\Theta$, yielding adapted parameters $\tilde{\Theta} = \Theta + \Delta\Theta$.

**Perplexity Objective.**  For token sequence $x = (x_1, \dots, x_T)$, perplexity is defined as:

$$\mathcal{P}(x; \Theta) = \exp\Big( -\frac{1}{T} \sum_{t=1}^{T} \log p(x_t \mid x_{1:t-1}; \Theta)\Big). \tag{6}$$

Since ground-truth answers $y$ are unavailable at test time, TTL minimizes *input* perplexity: $\min_\Theta \mathcal{P}(x; \Theta)$.

**Sample-Efficient Weighting.**  Not all test samples contribute equally. TTL uses a perplexity-based selection weight that prioritizes high-perplexity samples:

$$S(x) = \lambda \cdot \exp\big( \log \mathcal{P}(x; \Theta) - \log \mathcal{P}_0 \big)\, \mathbb{I}_{\{\mathcal{P}(x;\Theta) > \mathcal{P}_0\}}(x), \tag{7}$$

where $\mathcal{P}_0$ is a threshold and $\lambda$ is a scaling constant. Low-perplexity samples are excluded ($S(x) = 0$), focusing adaptation on samples where the model is most uncertain.

**LoRA-Based Adaptation.**  To prevent catastrophic forgetting and reduce compute, TTL updates only low-rank parameters $\Delta\Theta = \mathcal{B}\mathbf{A}$, yielding the weighted objective:

$$\min_{\Delta\Theta}\; S(x)\, \mathcal{P}(x; \Theta + \Delta\Theta). \tag{8}$$

**Configuration Schema.**

```
{
  ``family": ``TTT",
  ``ttl_steps": <integer>,
  // number of optimization steps
  ``learning_rate": <float>,
  // optimizer learning rate
  ``batch_size": <integer>,
  // samples per update
  ``shuffle_data": <boolean>
  // whether to shuffle test inputs
}
```

---

**Algorithm 2** TTL: Test-Time Learning via Weighted Perplexity Minimization

---

**Require:** Test batch $\mathcal{X} = \{x_b\}_{b=1}^{B}$, pretrained LLM $f_{\Theta}$, LoRA params $\Delta\Theta$, threshold $\mathcal{P}_0$
 1: Initialize LoRA: $\mathbf{A} \sim \mathcal{N}(0, \sigma^2)$, $\mathcal{B} = 0$
 2: $\tilde{\Theta} \leftarrow \Theta + \Delta\Theta$
 3: **for** each batch $\mathcal{X}$ **do**
 4:     Compute perplexities $\mathcal{P}(x_b; \tilde{\Theta})$
 5:     Compute weights $S(x_b)$; exclude low-perplexity samples
 6:     Update $\Delta\Theta$ by minimizing $\sum_{b=1}^{B} S(x_b)\,\mathcal{P}(x_b; \tilde{\Theta})$
 7: **end for**
 8: **Return:** adapted model $f_{\Theta + \Delta\Theta}$

---

## G.2 TWO-SUBSPACE MIXING LORA

This strategy enhances standard LoRA by enabling subspace interaction through a fixed butterfly mixing factor, providing richer representational capacity without additional trainable parameters.

**LoRA as Subspace Composition.** For frozen pretrained weights $\mathbf{W}_0 \in \mathbb{R}^{d_1 \times d_2}$ and input $x$, standard LoRA computes:

$$x\mathbf{W}_0 + x\Delta\mathbf{W} = x\mathbf{W}_0 + x\mathbf{A}\mathbf{B}, \tag{9}$$

with $\mathbf{A} \in \mathbb{R}^{d_1 \times r}$, $\mathbf{B} \in \mathbb{R}^{r \times d_2}$ and $r \ll \min(d_1, d_2)$. Decomposing into rank-1 components:

$$\mathbf{A} = [\mathbf{A}_1, \mathbf{A}_2, \ldots, \mathbf{A}_r], \quad \mathbf{B}^{\top} = [\mathbf{B}_1^{\top}, \mathbf{B}_2^{\top}, \ldots, \mathbf{B}_r^{\top}],$$

vanilla LoRA can be viewed as:

$$x\mathbf{A}\mathbf{B} = x\sum_{i=1}^{r} \mathbf{A}_i \mathbf{B}_i = x\mathbf{A}\mathbf{I}_{r \times r}\mathbf{B}. \tag{10}$$

**Two-Subspace Mixing.** The two-subspace mixing variant replaces the identity mixer with a butterfly factor that enables cross-subspace interaction:

$$x\sum_{i=1}^{r/2} (\mathbf{A}_i + \mathbf{A}_{i+r/2})(\mathbf{B}_i + \mathbf{B}_{i+r/2}) = x\mathbf{A} \begin{bmatrix} \mathbf{I} & \mathbf{I} \\ \mathbf{I} & \mathbf{I} \end{bmatrix} \mathbf{B}. \tag{11}$$

This formulation mixes information from $2r$ subspaces (compared to $r$ in vanilla LoRA), modeling richer interactions without introducing additional learnable weights.

**Interpolation Parameter.** The $\lambda$ hyperparameter controls the interpolation ratio between the two resulting subspace outputs during inference, allowing fine-grained control over the adaptation strength.

**Configuration Schema.**

```
{
  ``family": ``LORA",
  ``lambda": <float>
  // mixing ratio between subspaces
  // (0.0 to 1.0)
}
```

## G.3 TEST-TIME SCALING (TTS)

Test-Time Scaling leverages multiple prompt batches to improve prediction stability through either routing or ensembling. This approach is particularly effective for tasks where multiple plausible interpretations compete.

**Router Approach.** Given a test question, TTS samples $m$ prompt batches from the test split and selects the batch whose representation is closest to the question. Two routing methods are supported:

- **M1 (avg_sim_score):** Compute similarity scores between each prompt in a batch and the test question, then average across the batch. Select the batch with highest average similarity.
- **M2 (avg_prompt_embed):** Compute the mean prompt embedding for each batch, then select the batch whose mean embedding is closest to the test question embedding.

Euclidean distance was found to perform better empirically than cosine similarity for both methods.

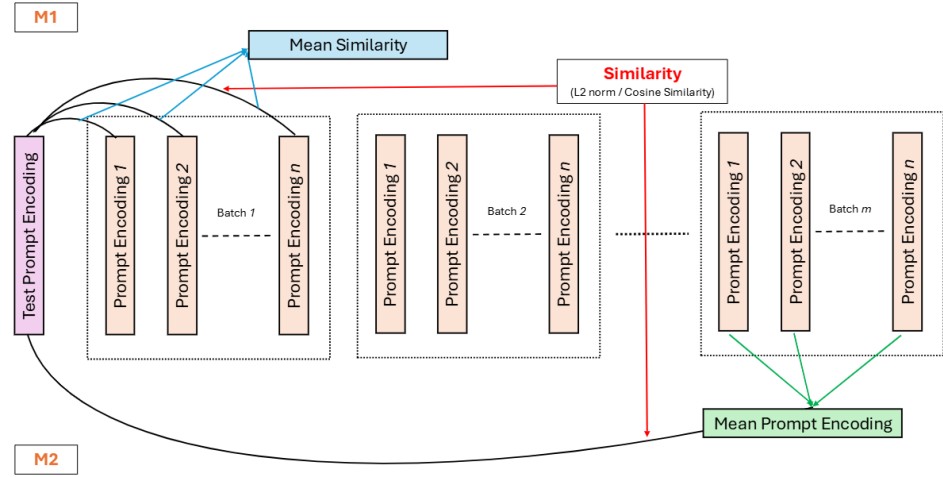

Figure 6: **TTS Router Architecture.** M1 computes mean similarity scores between the test prompt encoding and individual prompt encodings within each batch. M2 computes similarity between the test encoding and mean prompt embeddings per batch.

**Ensemble Approach.** For tasks requiring aggregation across multiple adapter configurations:

- **max_confidence:** Select the prediction with highest confidence score across all configurations.
- **majority_vote:** Aggregate predictions via voting across configurations.
- **sum_logprobs:** Sum log-probabilities across configurations for each candidate answer.

**Configuration Schema.**

```
{
  ``family": ``TTS",
  ``num_prompt_batches": <integer>,
  // batches sampled from test split
  ``method": ``<avg_sim_score |
  avg_prompt_embed |
  max_confidence |
  majority_vote |
  sum_logprobs>"
}
```

### G.4 LATENT SPACE MODIFICATION

We introduce a lightweight, per-sample parameter applied to the final hidden layer, enabling rapid adaptation without modifying backbone parameters.

---

**Algorithm 3** SLOT: Sample-Specific LM Optimization at Test-Time

---

1: **Input:** pretrained LM $\mathcal{M}(\theta)$, prompt $x$, steps $T$, lr $\eta$
2: Initialize $\delta \leftarrow 0 \in \mathbb{R}^{1 \times d}$
3: **for** $t = 0$ to $T - 1$ **do**
4:    $H \leftarrow \mathcal{M}_{\text{pre-LM}}(x)$    (can be cached)
5:    $\mathcal{L}(\delta) \leftarrow -\sum_{i=1}^{n-1} \log p(x_{i+1} \mid x_{1:i}, \delta)$
6:    $\delta \leftarrow \text{OptimizerStep}(\delta, \nabla_\delta \mathcal{L}; \eta)$
7: **end for**
8: **Decode:** generate with logits $W_{\text{LM}}(H_{\text{last}} + \delta)$
9: **Return:** generated continuation $y$

---

**Problem Setting.** Let $\mathcal{M}$ be a pretrained autoregressive LM with fixed parameters $\theta$. Given input prompt $x = (x_1, \ldots, x_n)$, the model produces final hidden features $H = \mathcal{M}_{\text{pre-LM}}(x) \in \mathbb{R}^{n \times d}$ before the LM head. Next-token probabilities are $p(y \mid x) = \text{softmax}(W_{\text{LM}} H)$.

**Sample-Specific Parameter.** SLOT introduces a per-sample additive parameter $\delta \in \mathbb{R}^{1 \times d}$, broadcast across sequence positions:

$$H' = H + \delta, \qquad \text{logits} = W_{\text{LM}}(H + \delta). \tag{12}$$

**Prompt-Stage Optimization.** Initialize $\delta^{(0)} = \mathbf{0}$ and optimize for $T$ steps by minimizing the negative log-likelihood of the prompt sequence:

$$\mathcal{L}(\delta) = -\sum_{i=1}^{n-1} \log p(x_{i+1} \mid x_{1:i}, \delta). \tag{13}$$

Since $\delta$ is applied only to the final hidden layer, features $H$ can be cached; each optimization step requires only forward/backward through the linear head $W_{\text{LM}}$, yielding negligible overhead.

**Generation Stage.** During autoregressive decoding, SLOT reuses $\delta_{\text{opt}}$ without further optimization:

$$x_{\text{next}} \sim \text{softmax}\big(W_{\text{LM}}(H_{\text{last}} + \delta_{\text{opt}})\big). \tag{14}$$

**Computational Cost.** Prompt-stage adaptation optimizes only $d$ parameters with per-step cost $O(d|V|)$ and generation-time overhead $O(d)$ per token.

**Configuration Schema.**

```
{
  ``family": ``Latent",
  ``slot_steps": <integer>,
  // optimization steps (T)
  ``slot_lr": <float>
  // learning rate (eta)
}
```

### G.5 STRATEGY SELECTION VIA REINFORCEMENT LEARNING

The refinement policy $\pi_\phi$ selects strategies through a language-model-driven generation procedure. Given task context and few-shot exemplars, the strategy model emits structured JSON outputs specifying both the selected family and its configuration. This ensures refinement policies remain interpretable and directly executable.

**Objective Formulation.** We first formulate the objective for outer-loop RL training which generates adaptation strategies $\alpha$. Let $\theta$ denote the parameters of the language model $\text{LM}_\theta$. In order to adapt to an unseen dataset (task) $\mathcal{D}$, DECOUPLED requires, $\tau$ which is a context containing information relevant

to the task and $\mathcal{E}$ which is the evaluation strategy and metric used to assess the model's downstream adaptation. Based on $\tau$, DECOUPLED generates an $\alpha$ and updates its parameters accordingly $\theta' \leftarrow \texttt{Update}(\theta, \alpha)$. We thus have an RL setup i.e., the model takes an *action* (generating $\alpha$), receives a *reward* $r$ based on $\texttt{LM}_{\theta'}$'s performance on $\mathcal{E}$ and updates its policy to maximize expected reward,

$$\mathcal{L}_{\text{RL}}(\theta_t) := -\mathbb{E}_{(\tau, \mathcal{E}) \sim \mathcal{D}} \left[ \mathbb{E}_{\alpha \sim \text{LM}_{\theta_t}(\cdot | \tau)} \left[ r(\alpha, \mathcal{E}, \theta_t) \right] \right]$$

It is to be noted that the reward assigned to a given action depends on the model parameters $\theta$ at the time the action is taken (since $\theta$ is updated to $\theta'$, which is then evaluated). An implication of this is that the while modeling the RL state, one must therefore include $\theta$ in the policy's parameters as well along with $\tau$, even though the policy's observation is limited to $\tau$ (because it is extremely infeasible to directly place $\theta$ in the LLM's context window). Therefore, the (state, action, reward) triples which have been collected by using an older model weights, $\theta_{\text{old}}$, will not be aligned for the current model $\theta_{\text{current}}$. Hence, an on-policy approach should be adapted, by which adaptation strategies are sampled from and, even more importantly, the rewards itself will be calculated using the current model.

In particular, the specific on-policy approach used is ReST$^{EM}$ Singh et al. (2024) where samples are first generated from the current model and are filtered by using binary feedback [$r(\alpha, \mathcal{E}, \theta_t)$ is 1 if on $\mathcal{E}$, $\alpha$ improves $\text{LM}_{\theta_t}$'s performance and is 0 otherwise]. The model is then fine-tuned on these samples and this continues in an iterative manner. Currently, only a deterministic number of samples are being generated, 20 to be precise. This could however be improvised to be dynamic in future version of the work wherein samples would continue to be generated until a particular confidence threshold, as determined by the model itself is reached instead. The same is true for number of iterations as well which is just 2 for now. The generation process employs a temperature of 1.0, nucleus sampling (`top-p = 0.9`) and top-k filtering (`top-k = 50`).

**Dataset-Specific Configurations.** Empirically discovered configurations exhibit intuitive alignment with task demands:

- **ARC-e, JAMA Clinical, PIQA → TTL:** Domain-specific or jargon-dense tasks benefit from token-level distribution correction via perplexity minimization. Example Configuration: {`ttl_steps: 25, learning_rate: 1e-5, batch_size: 4`}.
- **ARC-c, SocialIQA → Latent:** Abstract reasoning and social inference tasks require adjustment of hidden-state trajectories. Example Configuration: {`slot_steps: 5, slot_lr: 0.1`}.
- **BoolQ, GSM-MC, MATH-MC → LoRA:** Structured reasoning tasks benefit from subspace mixing. Example Configuration: {`lambda: 0.5`}.
- **HellaSwag, DivLogicEval, CodeMMLU → TTS:** Adversarial or multi-interpretation tasks benefit from ensemble-style aggregation. Example Configuration: {`num_prompt_batches: 20, method: max_confidence`}.

### G.6 JSON SCHEMAS AND PROMPTING TEMPLATE FOR STRATEGY GENERATION

For reproducibility, we provide the exact prompting template used to generate adaptation strategies. The model is instructed to output a single valid JSON object corresponding to one strategy family, with no additional text. Task context's are obtained using by prompting a foundational model with the few-shot examples of the unseen task analogous to Charakorn et al. (2025).

**System Prompt.**

```
You are an expert AI agent tasked with generating an optimal
adaptation strategy for a Large Language Model to improve its
performance on a new, unseen task. Your goal is to output a single,
structured JSON object that specifies the most promising strategy.
```

**User Prompt.**

```
<Task Context>

Your Instruction:
Based on the task context and method descriptions below, generate a
single JSON
object representing the most effective adaptation strategy. You must
choose one
strategy family and output strictly valid JSON with no extra text.

JSON Output Format \& Strategy Families:

(1) Test-Time Training (TTT)
{
  "family": "TTT",
  "ttl_steps": <integer>,
  "learning_rate": <float>,
  "batch_size": <integer>,
  "shuffle_data": <boolean>
}

(2) LoRA Modification (LORA)
{
  "family": "LORA",
  "lambda": <float>
}

(3) Test-Time Scaling (TTS)
{
  "family": "TTS",
  "num_prompt_batches": <integer>,
  "method": "<avg_sim_score | avg_prompt_embed | max_confidence |
  majority_vote | sum_logprobs>"
}

(4) Latent Space Modification (Latent)
{
  "family": "Latent",
  "slot_steps": <integer>,
  "slot_lr": <float>
}

Now, provide only the JSON object for the <task> dataset.
```

**Task Contexts**
*Generation Prompt*: This prompt is used for querying `GPT-4o mini` to obtain the task descriptions for datasets

```
You are given a small set of example question-answer pairs from an
unknown dataset. Your task is to infer the
underlying task definition and write a concise, professional task
description suitable for inclusion in a machine
learning benchmark paper.

Instructions:
1. Do not mention specific example questions or answers.
2. Infer the core objective, skills being evaluated and
type of reasoning required.
3. Describe what the model is expected to do and what
competencies are being tested.
4. Write in neutral, academic language.

Output a single self-contained task description paragraph.

Examples from the dataset are provided below:
```

```
<question-answer examples>

Output only the task description. Do not include analysis, bullet
points, or headings.
```

Hereby, are the example task contexts used in this work, **(a) ARC-c**

```
This task is about analyzing questions which examine your grasp of
scientific ideas. You must connect conceptual knowledge with practical
examples from geology, ecology and environmental changes. The
objective here is to evaluate various scientific scenarios and infer
the most logical explanations or definitions based on established
knowledge. This task will strengthen your analytical and reasoning
skills in the context of natural science. Your role is to interpret
questions focusing on earth science and biological interactions. This
demands a clear understanding of relevant processes, such as
decomposition, weathering and species adaptation.
```

## (b) ARC-e

```
Your job is to discern which information best answers a posed
question, focusing on practical examples and scientific principles.
This requires a strong grasp of underlying concepts in ecology or
physics. You will analyze questions that explore important connections
such as environmental issues or animal adaptations. Utilize your
background knowledge to evaluate and select the most fitting answer.
This task involves selecting answers that reflect accurate
relationships or effects seen in nature or society. You will need to
sort through potential choices critically to find the appropriate one.
```

## (c) HellaSwag

```
This task revolves around completing an unfinished text by selecting
an ending that matches its tone and context. It requires you to think
critically about how narratives develop and conclude effectively. This
task asks you to select a suitable conclusion for an unfinished
narrative or instructional content. It tests your comprehension and
reasoning skills as you assess how well each option aligns with the
given text. Your task involves completing an incomplete passage by
selecting the ending that logically continues the context provided.
This requires reading comprehension and the ability to infer meaning
from a text.
```

## (d) PIQA

```
You will explore practical questions and select an answer that
presents a logical and widely accepted approach to solve a given
problem or complete a task successfully. Analyze the provided
scenarios where practical advice or solutions are required, focusing
on selecting the most. commonly used or convenient method. Given a
question related to common tasks, your responsibility is to discern
which proposed solution aligns with typical practices or makes the
task easier to achieve.
```

## (e) WinoGrande

```
In this exercise, you need to read short narratives and discern which
person or object fits best within the context of the sentence.This
task requires synthesizing information from concise textual scenarios
to identify crucial elements that drive the narrative forward. The
goal is to evaluate descriptions and select the entity that best
aligns with the sentiments or actions presented in the scenario.
```

**(f) BoolQ**

```
Analyze the given details about various subjects, including movies,
sports and television shows. Your role is to confirm whether certain
claims are true or false. Your task is to determine the truthfulness
of specific statements based on the provided background information.
This requires careful reading and comprehension of the content. The
goal is to evaluate factual claims made in relation to highlighted
texts. You will need to discern whether the statements align with the
information provided.
```

**(g) GSM-8K**

```
You will be tasked with interpreting mathematical situations described
in words. The goal is to use logical
reasoning and calculations to determine the numerical answers based on
the context provided. This task challenges your problem-solving
abilities through mathematical reasoning. You must carefully read each
scenario and systematically work through the data to compute the final
outcome. Your role is to engage with practical math scenarios
presented as questions. The task requires translating textual data
into numerical operations that will lead you to the final solution.
```

**(h) MATH**

```
This task focuses on solving challenging mathematical problems that
require multi-step logical reasoning rather than direct formula
application. You will analyze competition-level mathematics questions
spanning topics such as algebra, geometry, number theory, probability
and calculus. The objective is to carefully interpret each problem,
identify appropriate problem-solving strategies and carry out precise
symbolic or numerical reasoning to arrive at a correct final answer.
This task emphasizes structured thinking, the use of mathematical
heuristics and the ability to connect multiple concepts within a
single solution. Your role is to reason through complex scenarios,
perform intermediate derivations when necessary and produce an exact
answer that adheres to standard mathematical conventions. The task
evaluates deep mathematical understanding and disciplined reasoning
rather than surface-level computation or pattern matching.
```

**(i) DivLogicEval**

```
This task focuses on evaluating your ability to perform precise
logical reasoning over natural language statements. You will be given
a set of premises written in fluent but often counterintuitive
language, where commonsense intuition alone may be misleading. Your
objective is to analyze the logical structure underlying these
statements and determine which option is logically entailed, not
entailed, or required as a missing assumption. The task demands
careful attention to implications, negations and dependencies between
statements rather than surface-level meaning. You must rely on formal
reasoning principles to assess whether conclusions follow from the
premises. This task is designed to isolate logical reasoning skills by
minimizing reliance on background knowledge or real-world
plausibility.
```

**(j) SocialIQA**

> This task focuses on reasoning about everyday social situations that
> involve human interactions, intentions and emotional responses. You
> are given a short context describing a social scenario, followed by a
> question that probes implicit social commonsense. Your objective is to
> select the most plausible answer from multiple choices based on how
> people typically think, feel, or act in such situations. The questions
> require you to infer motivations, emotional reactions, social norms,
> or likely actions before or after an event. This task evaluates your
> ability to reason about social dynamics, perspective-taking and
> cause-effect relationships in human behavior. Successfully completing
> this task demands an understanding of common social expectations and
> the ability to apply Theory of Mind reasoning to interpret the mental
> states of individuals involved.

### (k) CodeMMLU

> This task focuses on evaluating your understanding of programming
> concepts, code semantics and software reasoning across a wide range of
> difficulty levels. You will analyze questions that involve reading,
> interpreting and reasoning about code snippets written in common
> programming languages. The objective is to assess your ability to
> understand control flow, data structures, algorithms and
> language-specific behaviors. This task requires careful examination of
> program logic, identification of errors or expected outputs and
> selection of the most appropriate answer based on correct
> computational reasoning. Your role is to apply foundational and
> advanced coding knowledge to infer how programs execute and how
> modifications affect their behavior. The task emphasizes precise
> reasoning about syntax, semantics and program execution rather than
> surface-level pattern matching.

### (l) JAMA Clinical

> This task focuses on answering and explaining complex real-world
> clinical cases drawn from challenging medical scenarios. You are
> required to analyze detailed patient case descriptions that may
> include atypical presentations, incomplete information, or competing
> diagnoses. The objective is to apply clinical reasoning to identify
> the most appropriate diagnosis or next management step from multiple
> answer choices. Beyond selecting the correct option, this task
> emphasizes understanding why that choice is correct and why
> alternative options are less appropriate. Successfully completing this
> task requires synthesizing medical knowledge across domains,
> interpreting clinical findings and reasoning in a manner consistent
> with expert decision-making in real clinical settings. The task
> evaluates both diagnostic accuracy and the ability to justify medical
> decisions using coherent, medically sound explanations.

## H  BASELINE HYPER-PARAMETERS

For fair comparison, all baseline methods use the same LoRA configuration (rank $r = 8$ for 0.5B and $r = 16$ for 1.5B), training datasets and hardware setup as TEMPO-LLM. This section documents the specific hyperparameters used for each baseline method.

**MAML-en-LLM:**  Following the implementation in Sinha et al. (2024), we use LoRA adapters with rank $r = 8$ for the 0.5B model and $r = 16$ for the 1.5B model, with LoRA alpha $\alpha = 16$ and scaling factor $\alpha/r$ applied to weight updates. LoRA adapters are applied to the Q, V and O projections in attention layers. The method uses a MAML-2-1 configuration with $n = 1$ task per step and $k = 1$ adaptation step per task, trained with batch size 1 as specified in the original paper.

Both inner and outer learning rates are set to $1 \times 10^{-5}$ and training proceeds for 10 epochs with a maximum of 50,000 steps. The optimizer is a shared AdamW with $\beta_1 = 0.9$, $\beta_2 = 0.999$, $\epsilon = 10^{-8}$ and weight decay 0.01, where first and second moment estimates are shared between inner and outer optimizers. Gradient clipping is applied with maximum norm 1.0 and test-time adaptation uses 10 gradient steps on 25 support samples.

## I    LIMITATIONS AND FUTURE WORK

While TEMPO-LLM demonstrates consistent improvements over bi-level meta-learning baselines across multiple benchmarks, several limitations warrant acknowledgment and promising directions for future research emerge from this work.

**Model Scale and Architecture Coverage.**    Our experiments focus on Qwen2.5 models at 0.5B and 1.5B parameter scales. Validation on larger models (e.g., 7B, 13B, or 70B parameters) would strengthen claims regarding scalability and confirm whether the benefits of temporal separation persist at scales where models possess greater inherent capacity. Furthermore, extending evaluation to other LLM families beyond Qwen2.5-such as Llama Touvron et al. (2023), DeepSeek DeepSeek-AI et al. (2024), or Mixtral Jiang et al. (2024)-would provide evidence for the generality of TEMPO-LLM's design principles across architectural variations and pretraining regimes.

**Parameter-Efficient Fine-Tuning Method Generalization.**    TEMPO-LLM's current implementation relies exclusively on Low-Rank Adaptation (LoRA) as the parameter-efficient fine-tuning mechanism. Exploring how the temporal separation paradigm extends to alternative PEFT methods-such as prompt tuning Lester et al. (2021), prefix tuning Li & Liang (2021) or adapter layers Liu et al. (2022)-represents an important direction. Different PEFT methods impose distinct structural constraints on the adaptation process and understanding whether TEMPO-LLM's three-stage decomposition remains effective under these alternative parameterizations would clarify the scope of its applicability.

**Evaluation on Multi-Turn and Interactive Settings.**    Our evaluation is limited to single-turn benchmarks, where tasks can be evaluated through isolated question-answering or generation tasks. Multi-turn dialogue, interactive reasoning and task-oriented conversation settings introduce temporal dependencies, context accumulation and dynamic adaptation requirements that are not captured in our current experimental design. Extending TEMPO-LLM to these settings would require investigating how foundation consolidation, alignment and refinement interact with conversation state and whether stage boundaries should be reconsidered in temporally extended interactions.

**Identifying Task-Critical Parameters via Mechanistic Interpretability.**    Inspired by prior work in quantization, pruning and the lottery ticket hypothesis Tang et al. (2025), we observe that only a subset of parameters may be crucial for specific tasks. TEMPO-LLM currently updates low-rank parameters uniformly across all adapter layers. A promising future direction involves integrating principles from mechanistic interpretability to *first identify task-critical parameter subsets* before applying updates. This could involve analyzing circuit-level activations, gradient saliency, or attention attribution to selectively target parameters that contribute most to task-specific reasoning. Such an approach could further increase adaptation efficiency and sparsity beyond the improvement already observed.

**Unbounded Exploration of Full-Model Weight Modification.**    TEMPO-LLM operates within the constraint of frozen base model parameters, performing adaptation exclusively through LoRA adapters. A more challenging and open direction is enabling *unbounded exploration of the hypothesis space for full-model weight modification*. This would involve relaxing the frozen-backbone assumption and developing principled methods for selective full-parameter updates that preserve foundation knowledge while enabling deeper specialization. Potential approaches include learned masking strategies, dynamic parameter allocation, or hierarchical adaptation policies that determine which layers and parameters should be modified based on task characteristics.

**Continual and Sequential Adaptation.**    TEMPO-LLM's current formulation assumes a fixed training task distribution and one-time deployment to target tasks. Extending the framework to continual

learning settings-where tasks arrive sequentially and the model must adapt without catastrophic forgetting-poses interesting challenges. Questions include: (1) how frequently should foundation consolidation be re-executed as task distributions shift? (2) can alignment and refinement stages be made incremental? and (3) what role does temporal separation play in mitigating interference between sequentially learned tasks?

