# OpenReview forum: "Identifying Geometric Bottlenecks in Single-Stage Training: Observations from the Optimization Manifold"
_ICLR.cc/2026/Workshop/Sci4DL — Submitted to Sci4DL 2026_

### Official Review · Reviewer_2F4d · 2026-02-22

**Fit:** 2
**Significance:** 2
**Confidence:** 2

**Summary:**

The authors present TEMPO-LLM, a method that investigates why bi-level meta-learning methods often fail to adapt large language models (LLMs) to highly diverse task distributions.

**Strengths:**

I thought that the analysis of gradient interference were particularly interesting, and investigating methods that pull apart gradients from diverse tasks would be an interesting avenue for future work.

**Suggestions:**

While I believe the results are interesting, the paper is currently written in a way that obscures many of the interesting experiments the authors performed. It currently feels to me that almost all of the interesting results are presented in the appendix, with the main text of the paper reserved for nonspecific summaries of the results in the appendix. (It is a bit concerning that Figure 1 first appears in the appendix)

Given that this is a workshop submission, I did not have time to read through the entire appendix, but I would strongly suggest that the authors substantially reorganize the presentation. As things stand, I believe the ideas presented are interesting, but it is difficult to make a proper judgement of the authors' work.

---

### Official Review · Reviewer_EJY8 · 2026-02-27

**Fit:** 3
**Significance:** 2
**Confidence:** 2

**Summary:**

This paper compares two methods of LoRA fine-tuning in order to investigate whether gradient interference forces models to learn suboptimal solutions for both specific tasks and for generalizing to out-of-distribution tasks. They compare coupled optimization (MAML-en-LLM) to a method that temporally separates different stages of learning (TEMPO-LLM). They find that in most cases, TEMPO-LLM outperforms MAML-en-LLM, and they find that the LoRA adaptations for these two methods differ.

**Strengths:**

1.	Well-written and organized.
2.	The barcode-pattern plots in Figure 4 were interesting in showing the alignment of parameters in the LoRA updates across tasks. This does suggest that the temporally-spaced task information results in different adaptions.
3.	The significance of these claims, if true, is that optimization for continual learning should be performed carefully so that the model can learn specific tasks carefully without losing the ability to generalize.

**Suggestions:**

1.	I found it quite long for a workshop paper since all of the key results had to be found in the appendices. Every figure or table was in the appendix, with just the descriptions of the overall findings for the main text.
2.	Figure 3 did not convince me entirely of the authors’ claim that the temporally-separated optimization method (TEMPO-LLM) was superior to MAML-en-LLM. They claim that in Figure 3, the top plot has overlapping lines, indicating that the LoRA adaption was the same regardless of the task, while the bottom plot has non-overlapping lines, indicating that there are distinct LoRA adaptions for each task. However, the plots seem equally as overlapping versus not. This seems contradictory to Figure 4.

---

### Official Review · Reviewer_ziL8 · 2026-02-28

**Fit:** 2
**Significance:** 1
**Confidence:** 2

**Summary:**

The paper investigates optimization properties of meta-learning for LLMs, specifically comparing a MAML-style algorithm that couples cross-task consolidation and task-specific refinement into a single nested optimization against TEMPO-LLM, a proposed alternative that separates these objectives into sequential, temporally independent training stages. The central argument is that the temporal organization of the training process fundamentally shapes the optimization properties and influences the resulting model's adaptation capacity. This claim is supported empirically through gradient similarity analysis, LoRA weight sparsity measurements, visual inspection of per-task adaptation patterns, and stage ablations.

**Strengths:**

* The central research question is important for building effective multi-task models, and the paper's focus on how coupling versus decoupling objectives affects optimization geometry is an interesting new angle.
* The choice of gradient similarity, LoRA weight sparsity and visual adaptation patterns as comparison metrics is intuitive and well motivated for characterizing differences in optimization behavior between the two methods.
* The OOD task set is diverse and allows for a meaningful discussion of how results differ across tasks and what that reveals about the methods.

**Suggestions:**

* While the paper's focus on the effect of temporal organization of the training process is an interesting angle, the experimental design does not allow the results to support this hypothesis. Specifically, the claim that temporal organization is the sole difference between the two compared methods is not accurate. While MAML-en-LLM trains a single set of LoRA weights and performs few-shot task-specific adaptation at test time, TEMPO-LLM relies on several substantially different techniques, including a hyper-convolutional network, a VAE-based alignment stage and an RL-based adaptation scheme. Any of these components could explain the observed differences in optimization properties: the difference in gradient similarity and adaptation patterns could be a consequence of direct LoRA optimization versus hyper-network generation conditioned on task-specific prompts and difference in the adaptation methods, and the weight sparsity difference could naturally arise from the generative bottleneck in the hyper-network rather than from temporal staging itself. Therefore, while the empirical results do demonstrate differences between the two methods, it remains unclear whether these differences are attributable to temporal organization or to the many other architectural and methodological differences between the systems.
* The paper would benefit from clearer structure and more explicit description of key methodological details. In its current form, it is difficult to follow without carefully reading the appendix, as the main text does not adequately describe the building blocks of TEMPO-LLM, and all figures and tables with the main empirical results are moved to the appendix. Moreover, several important details remain unclear even after reading the appendix. First, the test-time adaptation procedure of the MAML baseline is not described clearly enough, which makes some experiments difficult to interpret. For instance, it is unclear whether adaptation is performed before computing gradients for the gradient similarity analysis. Also, the results for MAML in Figure 3 and Figure 4 seem inconsistent: Figure 4 shows LoRA weights as completely identical across tasks while the lines in Figure 3 do not overlap completely as the caption suggests. Second, the purpose and effect of the VAE-based alignment stage would benefit from a clearer explanation — it is not immediately obvious what "semantic alignment" means in this context, how exactly the weights are modified, and what the isolated contribution of this stage is to the final performance. A more fine-grained ablation comparing Stage 1, Stages 1+2 and the full pipeline would help address this.

---

### Meta-Review · Area_Chair_FHCp · 2026-03-01

**Recommendation:** Reject

**Metareview:**

I agree with the reviewers concern about the 'paper's focus on the effect of temporal organization of the training process is an interesting angle, the experimental design does not allow the results to support this hypothesis.' I also find the paper difficult to read and found certain claims in the paper not fully founded ( Fig 3, Fig 4). I think the paper would benefit from a rewrite and clearer organization with consistency in results.
I recommend rejecting the paper.

---

### Decision · Program_Chairs · 2026-03-02

Reject